# Diagnostic performance of deep learning in infectious keratitis: a systematic review and meta-analysis protocol

Zun Zheng Ong ,[1] Youssef Sadek,[1] Xiaoxuan Liu,[2] Riaz Qureshi,[3] Su-Hsun Liu ,[3] Tianjing Li,[3] Viknesh Sounderajah,[4,5] Hutan Ashrafian ,[4,5] Daniel Shu Wei Ting,[6,7] Dalia G Said,[1,8,9] Jodhbir S Mehta,[6,7] Matthew J Burton,[10,11] Harminder Singh Dua,[1,8] Darren Shu Jeng Ting [2,8,12]

For numbered affiliations see end of article.

**Correspondence to**
Dr Darren Shu Jeng Ting;
ting.darren@gmail.com

## ABSTRACT

**Introduction** Infectious keratitis (IK) represents the fifth-leading cause of blindness worldwide. A delay in diagnosis is often a major factor in progression to irreversible visual impairment and/or blindness from IK. The diagnostic challenge is further compounded by low microbiological culture yield, long turnaround time, poorly differentiated clinical features and polymicrobial infections. In recent years, deep learning (DL), a subfield of artificial intelligence, has rapidly emerged as a promising tool in assisting automated medical diagnosis, clinical triage and decision-making, and improving workflow efficiency in healthcare services. Recent studies have demonstrated the potential of using DL in assisting the diagnosis of IK, though the accuracy remains to be elucidated. This systematic review and meta-analysis aims to critically examine and compare the performance of various DL models with clinical experts and/or microbiological results (the current 'gold standard') in diagnosing IK, with an aim to inform practice on the clinical applicability and deployment of DL-assisted diagnostic models.

**Methods and analysis** This review will consider studies that included application of any DL models to diagnose patients with suspected IK, encompassing bacterial, fungal, protozoal and/or viral origins. We will search various electronic databases, including EMBASE and MEDLINE, and trial registries. There will be no restriction to the language and publication date. Two independent reviewers will assess the titles, abstracts and full-text articles. Extracted data will include details of each primary studies, including title, year of publication, authors, types of DL models used, populations, sample size, decision threshold and diagnostic performance. We will perform meta-analyses for the included primary studies when there are sufficient similarities in outcome reporting.

**Ethics and dissemination** No ethical approval is required for this systematic review. We plan to disseminate our findings via presentation/publication in a peer-reviewed journal.

**PROSPERO registration number** CRD42022348596.

### STRENGTHS AND LIMITATIONS OF THIS STUDY

⇒ This systematic review and meta-analysis will include all relevant articles related to the diagnostic performance of deep learning in infectious keratitis through a comprehensive search of all literature.

⇒ This review will examine the diagnostic performance of deep learning in identifying and differentiating the subtypes of infectious keratitis, including bacterial, fungal, viral and parasitic keratitis.

⇒ This review will examine the diagnostic accuracy of deep learning in distinguishing infectious keratitis from healthy eyes and other ocular surface diseases.

⇒ Meta-analysis will be performed to determine the pooled diagnostic sensitivity and specificity of deep learning and will be compared with the performance of healthcare professionals (if data are available).

⇒ The quality of the study will depend on the quality of the available published literature related to this topic.

## INTRODUCTION

Due to worldwide population ageing and urbanisation, it is expected that close to 900 million people will suffer from distance vision impairment, of whom 61 million people will be blind by 2050.[1] Infectious keratitis (IK), also commonly known as corneal infection, currently represents the fifth-leading cause of blindness worldwide.[2 3] It can be caused by a wide variety of pathogens such as bacteria, fungi, protozoa and viruses.[3 4] Once considered a 'silent epidemic' in low-income and middle-income countries (LMICs), IK has so far caused ~5 million cases of blindness around the world and is estimated to cause ~2 million monocular blindness each year, placing significant burden on global human health.[3 5] A recent meta-analysis conducted by Brown *et al*[6] estimated that the global incidence of fungal keratitis alone (without accounting for other types of IK) is likely >1 million cases per year, primarily affecting the populations in Africa and Asia. Previous studies have also consistently reported a disproportionately higher incidence of IK in the LMICs (113–799 per 100 000

populations-year) than in high-income countries (HICs; 2.5–40.3 per 100 000 populations-year),[3 7 8] which was likely attributable to increased risk of trauma from agricultural and other occupational activities, environmental factors, the use of traditional eye medicine (which may contain pathogens) and the limited access to primary and secondary eye care.[3 9–11]

Patients affected by IK are often debilitated by severe ocular pain and sight loss, and some are at risk of losing the eye due to intractable infection.[6 11–14] The outcome of IK is critically dependent on a timely and accurate diagnosis, followed by appropriate medical and/or surgical interventions. In current clinical practice, IK is usually diagnosed on clinical grounds with support from additional tests, including microbiological investigations (eg, smear microscopy, culture and sensitivity testing, and PCR) and/or corneal imaging (eg, in vivo confocal microscopy (IVCM)).[15–17] However, these approaches have multiple challenges, including the need for clinical expertise and equipment, low microbiological culture yield, long turnaround time, poorly differentiated clinical features and polymicrobial infections.[7 15 18–20] Moreover, access to such microbiological and imaging investigations is not available in many ophthalmic units in LMICs, leading to a reliance of empirical treatment. This can lead to a misdiagnosis when based on clinical features alone and the use of incorrect antimicrobial therapy (eg, fungal keratitis being treated only with antibacterial agents). This can result in delays in the initiation of effective treatment, with consequent poorer clinical outcomes and higher risk of ocular complications.

In recent years, the interest of integrating artificial intelligence (AI) into clinical medicine with the hope of improving the quality of healthcare services has been reignited,[21] primarily owing to the advancement in deep learning (DL) techniques, improvement in computing power and increased availability of big data.[22–26] DL, a subfield of AI, has demonstrated promise in assisting automated medical diagnosis, clinical triage and decision-making, as well as improving the workflow efficiency in healthcare services in both developed and developing countries.[23–25 27–30] Within the realm of ophthalmology, DL research previously focused mainly on various posterior segment diseases (eg, age-related macular degeneration, diabetic retinopathy and glaucoma) and demonstrated comparable, if not better, diagnostic accuracy compared with healthcare professionals.[23 24 31–33] Although several recent studies have demonstrated the potential of DL in assisting the diagnosis of IK and distinguishing IK from other ocular diseases,[34–39] the diagnostic accuracy of these DL models remains to be elucidated.

To the best of our knowledge, there is no published systematic review and/or meta-analysis specifically evaluating the diagnostic performance of DL in IK. In view of the current diagnostic challenges of IK and the potential of DL in addressing the highlighted limitations, this systematic review and meta-analysis aims to critically examine and compare the performance of various DL models with clinical experts (the current 'gold standard') in diagnosing IK, which can help inform the clinical practice on the potential clinical applicability and deployment of this DL model.

## Review questions/objectives
The proposed systematic review aims to answer the following main questions:
1. What is the diagnostic accuracy of DL models in detecting and differentiating IK from healthy eyes?
2. What is the diagnostic performance of DL models in differentiating IK from other types of corneal or ocular diseases?
3. What is the diagnostic accuracy of DL models in differentiating the types of IK (eg, bacterial keratitis vs fungal keratitis)?

## METHODS
This protocol was produced based on the Preferred Reporting Items for Systematic Reviews and Meta-Analyses Protocols (PRISMA).[40] This systematic review will be conducted in accordance with the recommendations of the Cochrane Handbook for Systematic Reviews of Diagnostic Test Accuracy (DTA). We will write the resulting paper following the PRISMA-DTA[41] and the CHecklist for critical Appraisal and data extraction for systematic Reviews of prediction Modelling Studies.[42]

### Eligibility criteria
This diagnostic accuracy systematic review will consider all relevant clinical studies, including prospective and retrospective comparative cohort studies, case–control studies and cross-sectional studies, which examined the accuracy of DL in diagnosing any types of IK, encompassing bacterial, fungal, *Acanthamoeba* and/or viral keratitis. We will exclude case reports and reviews. We will only include studies that employed corneal imaging tests, which may include slit-lamp photography, IVCM, anterior segment optical coherence tomography (AS-OCT) and/or corneal topography/tomography. We will exclude AI studies that contained only data without any imaging or those that focused on image segmentation instead of disease classification. There will be no restriction on patients' age, gender, ethnicity and geographical location. There will be no restriction on the number and proportion of images used for each stage of the DL models, including training, validation and testing stages.

### Information sources and search strategy
We will search various bibliographic databases, including EMBASE (OVID), MEDLINE (OVID), CINAHL and DANS EASY Archive, and trial registries, including the Cochrane Central Register of Controlled Trials (CENTRAL), ISRCTN registry (www.isrctn.com/), US National Institutes of Health Ongoing Trials Register (https://www.clinicaltrials.gov/) and WHO International Clinical Trials Registry Platform for primary research related to DL for diagnosing IK. The search strategy aims

**Table 1** A summary of the search strategy using EMBASE for studies related to artificial intelligence in diagnosing infectious keratitis

| # | Query | Results from 8 May 2022 |
|---|-------|------------------------|
| 1 | exp artificial intelligence/ | 60341 |
| 2 | artificial intelligence.mp. | 46874 |
| 3 | exp machine learning/ | 306312 |
| 4 | machine learning.mp. | 91038 |
| 5 | exp deep learning/ | 23936 |
| 6 | deep learning.mp. | 36705 |
| 7 | machine intelligence.mp. | 242 |
| 8 | exp support vector machine/ | 30564 |
| 9 | support vector machine.mp. | 34400 |
| 10 | computer-assisted.mp. | 943528 |
| 11 | visual data exploration.mp. | 18 |
| 12 | 1 or 2 or 3 or 4 or 5 or 6 or 7 or 8 or 9 or 10 or 11 | 1266088 |
| 13 | exp keratitis/ | 34942 |
| 14 | keratitis.mp. | 26696 |
| 15 | infectious keratitis.mp. | 1369 |
| 16 | infective keratitis.mp. | 130 |
| 17 | exp microbial keratitis/ | 8393 |
| 18 | microbial keratitis.mp. | 1606 |
| 19 | corneal infection.mp. | 1037 |
| 20 | exp cornea ulcer/ | 7787 |
| 21 | corneal ulcer.mp. | 2586 |
| 22 | 13 or 14 or 15 or 16 or 17 or 18 or 19 or 20 or 21 | 38271 |
| 23 | 12 and 22 | 681 |
| 24 | limit 23 to human | 595 |

to locate both published and unpublished studies. The search will be developed with two concepts built into the search strategy to capture relevant articles: (1) AI and (2) IK. There will be no restriction on the study design, date or language for the search. The search strategy, including all identified keywords and index terms, will be adapted to each included information source. The reference list of all eligible studies will be manually screened for additional studies. An example of the search strategy is provided in table 1.

## Study selection

Following the search, all identified citations will be loaded into EndNote V.X9 (Clarivate Analytics, Pennsylvania, USA) and duplicates will be removed. The titles and abstract will be screened by two independent reviewers for assessment against the inclusion criteria of the review, using the Rayyan AI platform (Qatar).[43] The full text of selected citations will then be assessed in detail against the inclusion criteria by two independent reviewers. Reasons

for exclusion of any full-text studies will be recorded and reported in the systematic review. Any disagreements between reviewers at each stage of study selection will be resolved through discussions or consultations with a third reviewer. The results of the search will be reported in full in the final systematic review and presented in a PRISMA-DTA flow diagram.[41]

## Data collection and data items

Data will be extracted from the included articles by two independent reviewers using a standardised and pilot-tested data extraction tool, RevMan V.5.4 (Copenhagen: The Nordic Cochrane Centre, Cochrane). The extracted data will include specific details about the name of authors, study title, year of publication, countries of study, populations (including diseased and healthy cases), demographic factors (ie, age, gender, ethnicity), sample size, study methods, types of DL algorithms, decision threshold, types of reference standard (which may include expert consensus, microbiological results confirmed on either smear microscopy, culture or PCR and/or corneal imaging such as IVCM), and diagnostic accuracy (including the sensitivity and specificity) of the index test (ie, DL algorithms) and the comparator (ie, non-expert healthcare professionals), if available. We will extract sufficient information to build 2×2 contingency tables at the reported threshold for each study. The contingency tables will include true positive, false positive, true negative and false negative to calculate the sensitivity and specificity. If various contingency tables were provided for the same or different algorithms in the same study, they are assumed to be independent from each other as the aim of this work is to provide an overview of the results of various studies instead of the precise point estimates.[30] Any disagreements will be resolved by group consensus. Authors of eligible studies will be contacted to request any missing data, where required.

## Outcomes

The primary outcome will be the diagnostic accuracy of DL algorithms in distinguishing IK from healthy eyes and/ or those with other types of corneal diseases, as compared with the reference standard. The diagnostic accuracy of each group will be presented in the form of sensitivity and specificity.[44] The secondary outcomes for this review will involve a comparison of the accuracy in differentiating various types of IK and in differentiating IK from other types of corneal or ocular surface diseases. For studies that focused on distinguishing IK (any type) from healthy corneas or those with other non-IK corneal pathologies, the reference standard will be the expert consensus and/ or microbiological results. For studies that focused on differentiating the subtypes of organisms (eg, bacteria vs fungi), the microbiological results or expert consensus (if microscopy or culture results were not available) will be the reference standard. Other potential outcomes for this review will include the accuracy of DL in predicting the culture positivity and clinical outcomes of IK based on the

initial presenting images. These secondary outcomes may not be feasible if these more specific questions are not ascertained in the included studies, but they are nonetheless of interest.

## Risk of bias assessment

Eligible studies will be critically appraised by two independent reviewers at the outcome level for methodological quality in the review using the Quality Assessment of Diagnostic Accuracy Studies-2 (QUADAS-2) tool.[45] QUADAS-AI (an AI-specific extension to QUADAS-2) tool[46] will be used if available by the time of the conduct of this systematic review. Specifically, we will assess the Risk of Bias for our primary outcomes (ie, accuracy of DL vs reference standard for IK). The questions used in these tools are split into four domains: patient selection, index test, reference test, and flow and timing. Each of these domain help assess the risk of bias created by patient selection, the conduct and interpretation of index test and reference test and the sequence and timing of the study, respectively. We will also assess whether the AI systems have been tested on an externally validated test set. Authors of papers will be contacted to request for additional data for clarification, where necessary. Any disagreements will be resolved by discussion or by seeking the advice of a third reviewer. All studies, regardless of the results of their methodological quality, will undergo data extraction and synthesis. The results of critical appraisal will be reported in the final systematic review, in both narrative and tabular formats.

## Data synthesis and analysis

The analysis will be conducted at two levels: (1) a systematic synthesis of all eligible studies and (2) a meta-analysis of all relevant studies with similar outcome reporting. For the meta-analysis, the intervention group (ie, the 'index test') will refer to the image-based DL algorithms for diagnosing or differentiating IK from other ocular diseases. The reference group will be the expert consensus and/or microbiological results, also known as the gold standard or 'ground truth' for the DL algorithms whereas the comparator group, if available, will be the non-expert healthcare professionals.

Where possible, we will pool similar measures of accuracy from studies with statistical meta-analysis using RevMan V.5.4 software. 'Paired' forest plots, with one forest plot for sensitivity and the other for specificity, will be used and presented side by side.[47] The means and 95% CIs of each selected primary studies will be presented alongside the number of true positives, false positives, true negatives and false negatives, wherever appropriate. Summary receiver operating characteristic (SROC) curves will also be plotted using the sensitivity and specificity of each primary study. $\chi^2$ or Fisher's exact tests will be used to assess the heterogeneity objectively, if needed.[47] We expect heterogeneity in the types of DL systems and algorithms used across studies and we will consider all to be acceptable 'interventions' for analysis as

our question is meant to assess the general accuracy of any DL system. In view of the anticipated interstudies heterogeneity, a random-effects model will be used for the meta-analysis to determine the pooled sensitivity and specificity of the included studies. A fixed-effect model may be used if there is no significant heterogeneity. Subgroup analyses will be conducted where there are sufficient data to investigate different types of IK and other ocular surface diseases, as per our prespecified secondary outcomes. Additional subgroup analysis will also be performed on the diagnostic performance of the DL systems based on different imaging modalities, including slit-lamp photography, IVCM, AS-OCT and others.

The heterogeneity between studies will be assessed through the graphic display of paired forest plots or SROC curves. We will evaluate potential publication bias of the pooled data using Deek's funnel plot, and a $p < 0.05$ will be considered of significant publication bias.[48 49] The Grading of Recommendations, Assessment, Development and Evaluation approach for grading the certainty of evidence will be followed.[40] A Summary of Findings (SoF) table, created using GRADEpro software, will be presented. Where appropriate, the following information will be included in the SoF: number and type(s) of studies contributing to the outcome, total sample size contributing to the outcome, ranking of the certainty of the evidence based on the risk of bias, heterogeneity, directness, publication bias and precision of the review results. We will include the following outcomes in the SoF table: AUC, sensitivity and specificity for IK overall (ie, primary outcome).

## Patient and public involvement

DSJT had previously involved patients who were affected by IK to help identify the research need and priority in relation to IK. Many of the patients with IK have highlighted the importance of timely and accurate diagnosis of IK as the delay in diagnosis has negatively affected their visual outcomes. This serves as one of the key reasons for conducting this systematic review and meta-analysis, which aims to improve the diagnosis of IK in clinical settings.

## Clinical relevance of this systematic review

The results of this systematic review and meta-analysis will provide high-quality evidence on the diagnostic accuracy of DL in IK. This study will help identify the gaps (if any) in the current clinical evidence, which may be related to study design, quality of the research methodologies, setting of reference standard, risk of bias and outcome reporting. The identification of these issues can help refine the study design of any future clinical trials evaluating the diagnostic accuracy of DL in IK in a real-world setting. These findings will also help inform the clinicians, researchers, policy-makers and regulatory bodies on the clinical applicability of DL in diagnosing IK, with an aim to develop more accessible investigations for IK in the future, including in both HIC and LMICs.

**Author affiliations**
[1]Department of Ophthalmology, Queen's Medical Centre, Nottingham, UK
[2]Academic Unit of Ophthalmology, Institute of Inflammation and Ageing, University of Birmingham, Birmingham, UK
[3]Department of Ophthalmology, School of Medicine, University of Colorado Anschutz Medical Campus, Aurora, Colorado, USA
[4]Institute of Global Health Innovation, Imperial College London, London, UK
[5]Department of Surgery & Cancer, Imperial College London, London, UK
[6]Duke-NUS Medical School, National University of Singapore, Singapore
[7]Singapore National Eye Centre, Singapore Eye Research Institute, Singapore
[8]Academic Ophthalmology, School of Medicine, University of Nottingham, Nottingham, UK
[9]Research Institute of Ophthalmology, Cairo, Egypt
[10]International Centre for Eye Health, London School of Hygiene and Tropical Medicine, London, UK
[11]National Institute for Health Research (NIHR) Biomedical Research Centre (BRC) for Ophthalmology, Moorfields Eye Hospital NHS Foundation Trust and UCL Institute of Ophthalmology, London, UK
[12]Birmingham and Midland Eye Centre, Birmingham, UK

**Contributors** DSJT designed and conceptualised the study. ZZO, YS, XL, RQ, S-HL, TL, VS, HA, DT, DS, JM, MJB, HSD and DSJT developed the study protocol. ZZO and YS collected the data. ZZO, YS, XL, RQ, S-HL, TL, VS, HA, DT, DS, JM, MJB, HSD and DSJT interpreted the data. ZZO and DSJT drafted the initial manuscript. ZZO, YS, XL, RQ, S-HL, TL, VS, HA, DT, DS, JM, MJB, HSD and DSJT critically reviewed, revised and approved the final manuscript. DSJT acts as the guarantor of the study.

**Funding** MJB is supported by the Wellcome Trust (grant number 207472/Z/17/Z). DSJT is supported by Medical Research Council/Fight for Sight Clinical Research Fellowship (MR/T001674/1).

**Disclaimer** The funder has no role in developing this protocol.

**Competing interests** None declared.

**Patient and public involvement** Patients and/or the public were not involved in the design, or conduct, or reporting, or dissemination plans of this research.

**Patient consent for publication** Not applicable.

**Provenance and peer review** Not commissioned; externally peer reviewed.

**ORCID iDs**
Zun Zheng Ong http://orcid.org/0000-0002-3091-1871
Su-Hsun Liu http://orcid.org/0000-0002-1151-7983
Hutan Ashrafian http://orcid.org/0000-0003-1668-0672
Darren Shu Jeng Ting http://orcid.org/0000-0003-1081-1141

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
