## [Reviewer comments · BMJ Open]

ARTICLE DETAILS

TITLE (PROVISIONAL)	Diagnostic Performance of Deep Learning in Infectious Keratitis: A Systematic Review and Meta-Analysis Protocol
AUTHORS	Ong, Zun Zheng; Sadek, Youssef; Liu, Xiaoxuan; Qureshi, Riaz; Liu, Su-Hsun; Li, Tianjing; Sounderajah, Viknesh; Ashrafian, Hutan; Ting, Daniel; Said, Dalia; Mehta, Jodhbir; Burton, Matthew J; Dua, Harminder; Ting, Darren Shu Jeng

VERSION 1 – REVIEW

REVIEWER	Kuo, Ming-Tse Chang Gung Memorial Hospital Kaohsiung Branch, Ophthalmology
REVIEW RETURNED	03-Jul-2022

GENERAL COMMENTS	The author aims to perform a systematic review and meta-analysis for critically examining and comparing the performance of various deep learning models with clinical experts and/or microbiological results in diagnosing infectious keratitis, with an aim to inform practice on the clinical applicability and deployment of deep learning-assisted diagnostic models. However, 1. The author did not really report the result of their systemic review. 2. The author did not cite eligible references for this topic. There are about 1/4 references with the same first author (Darren S. J. Ting), the corresponding author of this article.
---

REVIEWER	Natarajan, Radhika Sankara Nethralaya, Medical Research Foundation, Deputy Director, Department of Cornea & Refractive Surgery
REVIEW RETURNED	02-Nov-2022

GENERAL COMMENTS	Commendable effort of collating articles on the use of Deep Learning for Infectious Keratitis. Will await the outcome article.
--

REVIEWER	Cabrera-Aguas, Maria The University of Sydney
REVIEW RETURNED	28-Nov-2022

GENERAL COMMENTS	Methods • Strategies: search in opengrey, CINAHL, metaRegister of controlled trials database • In your queries, add terms such as bacterial keratitis, HSV (herpetic) keratitis, HZV keratitis, fungal keratitis, acanthamoeba keratitis along with deep learning, etc
--

REVIEWER	Ye, Juan
-----------------	----------

	Zhejiang University, Department of Ophthalmology
REVIEW RETURNED	20-Feb-2023

GENERAL COMMENTS	Thanks for presenting this protocol. This ongoing study concentrated on the systematic review and meta-analysis of deep learning systems for infectious keratitis diagnosis. The search strategy in the protocol provides comprehensive keywords and therefore is of high reference value. But, there is a concern: on page 7, eligibility criteria, it is stated that you would include studies with different imaging modalities such as "slit-lamp photography," "AS-OCT," and so on, but there are no further discussions about this factor in the rest of the content. Imaging modalities should be taken into account when conducting your analysis because it is a crucial factor in studies involving deep learning.
--

VERSION 1 – AUTHOR RESPONSE

Reviewer 1

Comments to the Author:

The author aims to perform a systematic review and meta-analysis for critically examining and comparing the performance of various deep learning models with clinical experts and/or microbiological results in diagnosing infectious keratitis, with an aim to inform practice on the clinical applicability and deployment of deep learning-assisted diagnostic models.

Comment 1. However, the author did not really report the result of their systemic review.

Response: We thank the reviewer for the comment. As this is a systematic review protocol, it will not contain any results. However, we aim to complete the actual systematic review in the near future and publish the work in a peer-reviewed journal.

Comment 2. The author did not cite eligible references for this topic. There are about 1/4 references with the same first author (Darren S. J. Ting), the corresponding author of this article.

Response: As this is a systematic review protocol and not the actual systematic review, we have not included the full list of relevant deep learning studies for infectious keratitis in this protocol (since the protocol is developed before the comprehensive search for all relevant literature). However, as advised by the reviewer, we have included some additional relevant deep learning studies in this protocol, including #Ref 36 (PMID: 34145294) and #Ref 37 (PMID: 34359329), and will ensure to include all relevant articles in the full systematic review and meta-analysis. In addition, all the articles included is relevant to this article (including those published by the corresponding author).

Reviewer 2

Comments to the Author:

Comment 1. Commendable effort of collating articles on the use of Deep Learning for Infectious Keratitis. Will await the outcome article.

Response: We thank the reviewer for the positive comment. We aim to complete the proposed systematic review over the next few months and publish the results in a peer-reviewed journal.

Reviewer 3

Comments to the Author:

Comment 1. Methods - Strategies: search in opengrey, CINAHL, metaRegister of controlled trials database

Response: As recommended, we have included CINAHL as another information source but we believe opengrey and metaRegister of controlled trials (mRCT) database are no longer active since 2010 and 2014, respectively. We will search DANS EASY Archive (which contains opengrey archived articles) and ISCTRN and clinicaltrials.gov (which both contain all relevant articles in mRCT). Please see Page 8 for the updated "Information sources and search strategy" section.

Comment 2. In your queries, add terms such as bacterial keratitis, HSV (herpetic) keratitis, HZV keratitis, fungal keratitis, acanthamoeba keratitis along with deep learning, etc

Response: In our search strategy, we have used and included "keratitis" as the broader term to ensure we have captured all types of keratitis, including bacterial keratitis, fungal keratitis, viral keratitis, parasitic / Acanthamoeba keratitis, and other types of keratitis.

Reviewer 4

Comments to the Author:

Comment 1. Thanks for presenting this protocol. This ongoing study concentrated on the systematic review and meta-analysis of deep learning systems for infectious keratitis diagnosis. The search strategy in the protocol provides comprehensive keywords and therefore is of high reference value. But there is a concern: on page 7, eligibility criteria, it is stated that you would include studies with different imaging modalities such as "slit-lamp photography," "AS-OCT," and so on, but there are no further discussions about this factor in the rest of the content. Imaging modalities should be taken into account when conducting your analysis because it is a crucial factor in studies involving deep learning.

Response: We thank the reviewer for the constructive comment. As advised, we will perform subgroup analysis on the diagnostic performance of the deep learning systems based on different imaging modalities, including slit-lamp photography, in vivo confocal microscopy, and anterior segment optical coherence tomography. Please see Page 12 for the updated “Data synthesis and analysis” section.